# BMDD: A probabilistic framework for accurate imputation of zero-inflated microbiome sequencing data

Huijuan Zhou[1], Jun Chen[2]*, Xianyang Zhang[3]*

**1** School of Statistics and Data Science, Shanghai University of Finance and Economics, Shanghai, China, **2** Department of Quantitative Health Sciences, Mayo Clinic, Rochester, Minnesota, United States of America, **3** Department of Statistics, Texas A&M University, College Station, Texas, United States of America

* chen.jun2@mayo.edu (JC); zhangxiany@stat.tamu.edu (XZ)

## Abstract

Microbiome sequencing data are inherently sparse and compositional, with excessive zeros arising from biological absence or insufficient sampling. These zeros pose significant challenges for downstream analyses, particularly those that require log-transformation. We introduce BMDD (BiModal Dirichlet Distribution), a novel probabilistic modeling framework for accurate imputation of microbiome sequencing data. Unlike existing imputation approaches that assume unimodal abundance, BMDD captures the bimodal abundance distribution of the taxa via a mixture of Dirichlet priors. It uses variational inference and a scalable expectation-maximization algorithm for efficient imputation. Through simulations and real microbiome datasets, we demonstrate that BMDD outperforms competing methods in reconstructing true abundances and improves the performance of differential abundance analysis. Through multiple posterior samples, BMDD enables robust inference by accounting for uncertainty in zero imputation. Our method offers a principled and computationally efficient solution for analyzing high-dimensional, zero-inflated microbiome sequencing data and is broadly applicable in microbial biomarker discovery and host-microbiome interaction studies.

## Author summary

Understanding the microbes living in and on our bodies—the microbiome—relies on analyzing complex sequencing data. However, these data often contain many zeros, either because a microbe is truly absent or simply missed due to insufficient sampling. These missing values make it hard to accurately analyze microbial patterns and identify important differences between groups, especially for methods that work on a log scale. To address this, we developed a new method called BMDD that uses a more realistic model to impute the zeros. Unlike existing tools that assume each microbe follows an

**Data availability statement:** The Supplementary Materials provide the technical details and additional application results

referenced in the main text. The BMDD package is available at GitHub (https://github.com/zhouhj1994/BMDD) and CRAN (https://CRAN.R-project.org/package=MicrobiomeStat). The entire code and data for generating the presented results are available at GitHub (https://github.com/zhouhj1994/BMDD-manuscript-sourcecode).

**Funding:** This work was supported by the National Institutes of Health (R01GM144351 to JC & XZ); the National Science Foundation (DMS-1830392, DMS-2113359, DMS-1811747 to XZ; DMS-2113360 to JC); and the Mayo Clinic Center for Individualized Medicine (JC). The funders had no role in study design, data collection and analysis, decision to publish, or preparation of the manuscript. The primary work was conducted in the United States with NIH/NSF support. After returning to China, HZ completed additional work using non-U.S. funds; no NIH funds supported work performed outside the United States.

**Competing interests:** The authors have declared that no competing interests exist.

unimodal abundance distribution, BMDD allows for microbes to follow a bimodal distribution, so they could behave differently in different conditions. It provides not just a single guess, but a range of possible values to better reflect the uncertainty. Our testing shows that BMDD more accurately recovers the true microbial profiles and improves the ability to detect meaningful differences between groups. This method can help researchers gain clearer insights into how the microbiome affects health and disease.

## 1. Introduction

The human microbiome has been increasingly recognized as playing a critical role in health and disease [1–7]. For example, immune maturation and modulation [8,9], inflammatory cytokine production [10], host serum metabolome and insulin level [11], and host gene regulation [12] have all been shown to be linked to the human microbiome. Complementary to existing omics data, microbiome data offers an additional perspective on human health [1]. The dysbiosis of the human microbiome has been implicated in numerous human diseases [13].

Next-generation sequencing allows for the determination of the microbiome composition by direct microbial DNA sequencing [14]. Microbiome taxonomic abundance data, usually in the form of a count table of detected taxa, are typically over-dispersed and sparse with many zeros. Zeros could be due to either the physical absence of the taxa or their low abundance so that the sequencing machine cannot detect them reliably. Excessive zeros pose many statistical challenges for microbiome data analysis [15] and are particularly problematic for logarithmic scale analysis. The traditional approach involves adding a pseudocount, such as 1 or 0.5, to all counts before normalizing the data into relative abundances. Downstream statistical analyses are then based on the relative abundance data (a.k.a. compositional data), which sum up to one for each sample. Adding a pseudocount has been shown to have detrimental consequences in certain contexts [16].

The pseudocount approach, as a naive method to impute the zeros, does not fully exploit the information in the data and thus is far from optimal. An efficient zero-imputation approach should be able to tap into the correlation structure and the distributional characteristics of the data. To improve over the pseudocount approach, various zero-imputation techniques have been proposed for zero-inflated count data, including single-cell RNA sequencing (scRNA-seq) and microbiome sequencing data. One popular approach is to assume an underlying true abundance or expression matrix and estimate it using Bayesian methods. Examples of such methods include SAVER [17] for scRNA-seq data and mbDenoise [18] for microbiome data. SAVER models the count using a Poisson distribution, where the mean is the product of the true gene expression level and a normalization factor. A gamma prior is imposed on this true expression level. The hyperparameters in the priors are empirically estimated by fitting linear regression models between each gene and other genes in the same cell. On the other hand, mbDenoise uses zero-inflated negative binomial (ZINB) distributions to model taxon abundances. The mean of the negative binomial distribution is represented by the low-rank approximation of the count matrix. The posterior mean of the ZINB model is estimated using variational approximation and is used to recover the true abundance level. Another approach to microbiome data imputation, mbImpute [19], assumes that each taxon's abundance follows a mixture model: a gamma distribution for missing abundances requiring imputation and a log-normal distribution for actual (non-missing) abundances. For abundance values identified as non-missing, mbImpute fits linear models that combine information from similar samples, similar taxa, and sample covariates, then imputes

the missing values by the fitted values. The authors of mbImpute also proposed a method called scImpute for imputing scRNA-seq data employing principles similar to those used in mbImpute [20]. ALRA is another method for imputing scRNA-seq data [21], which uses singular value decomposition for low-rank approximation of the expression matrix. The rank is determined by identifying noise singular values, and data below a specified quantile after low-rank approximation is thresholded to preserve biological zeros.

Existing imputation methods either explicitly or implicitly assume a feature-wise unimodal distribution of the nonzero proportions/counts, which may be too restrictive for real microbiome abundance data. Here, we propose a more general Bayesian approach to impute zeros in microbiome abundance data using the posterior samples of the underlying true composition. The method is based on a structured prior that has better modeling capability than its predecessors in capturing the essential distributional characteristics of the composition data. Specifically, we propose a BiModal Dirichlet Distribution (BMDD) to model the prior distribution of the true composition. BMDD assumes that each component of the compositional vector follows a bimodal distribution, which is motivated by the observation that some taxa exhibit two modes in their abundance distributions [22]. The bimodal distribution could also result from a specific sampling scheme, such as a case-control design, where the cases and controls have different distributions. Moreover, excessive zeros could be efficiently modeled by using a mode sufficiently close to 0 in BMDD. Fig 1a shows the fit of BMDD to the abundance data of four example taxa from the American Gut Project [23], with taxa compositions estimated as posterior means obtained via the variational approximation (see Sections 4.2 and 4.3 for details). We can see that BMDD provides a better fit than the unimodal Dirichlet distribution. Note that BMDD can also model unimodal distributions and provides a more robust fit to unimodal taxa (see, for example, the top two taxa in Fig 1a) due to its increased modeling capability using more parameters.

Our method involves two major steps to find the posterior distribution: (i) we use a mean-field approach to approximate the form of the posterior, which is computationally intractable; (ii) we develop a variational EM algorithm to estimate the hyperparameters in BMDD. The posterior means obtained can be used as estimates of the true compositions. We conducted extensive simulation studies to evaluate the performance of different methods in estimating the true compositions. The results show that BMDD outperforms not only under the correctly specified model but also under misspecified and nonparametric models and thus could be beneficial for downstream statistical tasks such as differential abundance analysis, clustering, and prediction. Based on our model, we propose a multiple imputation approach for differential abundance analysis and demonstrate that it enhances the robustness of log-linear models by accounting for more uncertainty.

## 2. Results

### 2.1. Overview of the BMDD method

Our method is motivated by the observation that some taxa exhibit bimodal abundance distributions (for example, see the bottom two taxa in Fig 1a). One popular model for compositional data, which has a sum-to-one constraint, is the Dirichlet distribution (DD) [24]. However, DD is not capable of modeling the bimodal distribution observed for some taxa. To address DD's limitation, we propose the Bimodal Dirichlet Distribution (BMDD) to model the microbiome composition data more flexibly. The BMDD will then be used as the prior distribution under the empirical Bayes framework for zero imputation.

To understand BMDD, we first state a basic fact regarding the Dirichlet distribution. Let $Y_j$ be generated independently of the Gamma distribution with the shape parameter $\alpha_j > 0$ and

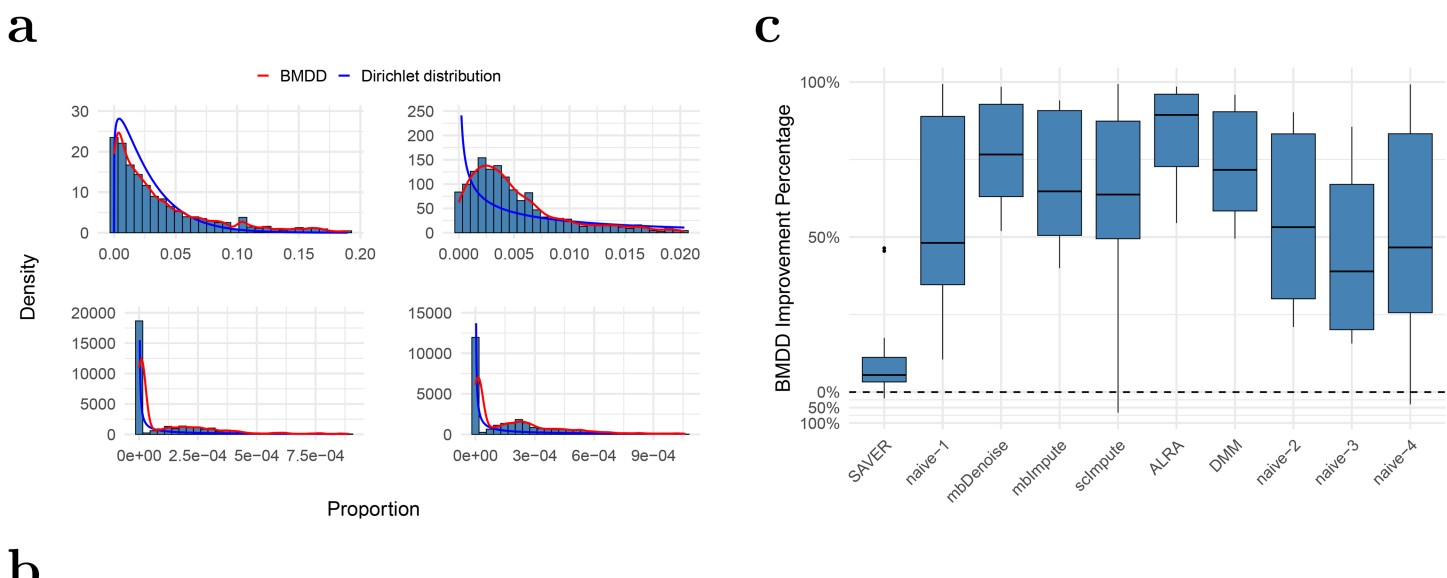

**Fig 1. Model fit and imputation performance of BMDD. a**: BMDD fit on a real human gut microbiome dataset from the American Gut Project. The blue curve shows the marginal density of the posterior Dirichlet distribution obtained from fitting the Dirichlet-multinomial model, and the red curve depicts the kernel density estimate based on the posterior means obtained from the fit of BMDD. **b**: Overview of the BMDD. The upper part shows the data generation mechanism of BMDD, and the lower part shows the model computation and inference. Deeper colors indicate higher abundances, and blanks indicate zeros. Details can be found in Section 2.1. **c**: Imputation performance of BMDD compared with competing methods across 15 distance metrics between the estimated and true composition matrices under the simulation setting S1. The performance improvement is expressed as the percentage reduction in distance metrics relative to the respective method. Formula for the values above zero (the dashed horizontal line): $\frac{\text{SAVER}-\text{BMDD}}{\text{SAVER}}$, the distance reduction of BMDD relative to SAVER; Formula for the values below zero $\frac{\text{BMDD}-\text{SAVER}}{\text{BMDD}}$, the distance reduction of SAVER relative to BMDD. Likewise for other competing methods.

the scale parameter $\rho > 0$, that is, Gamma$(\alpha_j, \rho)$ for $1 \leq j \leq m$. Setting $V = \sum_{j=1}^{m} Y_j$, we have $(X_1, \ldots, X_m) := (Y_1/V, \ldots, Y_m/V) \sim$ Dirichlet $(\alpha_1, \ldots, \alpha_m)$. $Y_j$, $X_j$ and $V$ can be interpreted as the absolute abundance for taxon $j$, relative abundance for taxon $j$ and the total microbial load, respectively. Motivated by this basic fact and the intention to capture the bimodal shape of taxa distribution, we consider a two-component mixture distribution for each $Y_j$, that is,

$$Y_j | \delta_j \sim (1 - \delta_j)\text{Gamma}(\alpha_{j,0}, \rho) + \delta_j\text{Gamma}(\alpha_{j,1}, \rho),$$

where $\delta_j \sim$ Bernoulli $(\pi_j)$ independently over $j$, with success probability $0 \leq \pi_j < 1$. Conditional on $(\delta_1, \ldots, \delta_m)$, one has

$$(X_1, \ldots, X_m) \sim \text{Dirichlet}(\alpha_{1, \delta_1}, \ldots, \alpha_{m, \delta_m}),$$

which is independent of the choice of $\rho$. Fig 1b illustrates the BMDD data generation process. Each taxon is parameterized by $(\alpha_{j,0}, \alpha_{j,1}, \pi_j)$, reflecting the mean absolute abundance for the two modes and the probability of the second mode. In Fig 1b(1), we show three representative taxa: one with near zero $\alpha_{j,0}$ (bimodal with the spike near 0), one with $\alpha_{j,0}, \alpha_{j,1}$ far apart (bimodal) and one with $\alpha_{j,0}, \alpha_{j,1}$ in proximity (unimodal). Next, based on the parameter $\pi_j$, we determine the mode from which the taxon comes for each sample (Fig 1b(2), '0' and '1' represent the two modes). With the mode assignment for each taxon, we generate the true composition (relative abundances) using the Dirichlet distribution (Fig 1b(3)). Note that taxon 1 has many relative abundances near 0. Given the composition and the sequencing depth, we generate the observed counts using a multinomial distribution (Fig 1b(4)). Now, taxon 1 has excessive zeros due to extremely low abundances in most samples.

Under this model, the posterior density of the true composition is computationally intractable because there are $2^m$ possible combinations of $\delta_j$'s (see Section 4.3 for details), so we use a variational inference approach to approximate the posterior. The hyperparameter of the model is estimated using variational EM. The true compositions are then estimated using the (approximate) posterior mean (Fig 1b(5)). We use a posterior predictive check to demonstrate that our method accurately recapitulates the observed data. Specifically, using data from the American Gut Project [23] as an example, we generate a count matrix by drawing samples from multinomial distributions parameterized with posterior samples of the compositions from BMDD. As shown in Fig A1 of the S1 File for the four example taxa in Fig 1a, the generated data closely resemble the distribution of the observed data.

The estimated posterior mean can be used as the imputed composition for downstream statistical analysis and machine learning tasks. Furthermore, we can generate multiple posterior samples from the posterior distribution (Fig 1b(6)). The multiple imputed compositions account for the estimation uncertainty, and the results on individual imputed datasets can be combined to improve the robustness of the analysis using Rubin's rule [25] or more specialized methods (see Section 2.3 for the example).

**Remark 1.** BMDD differs fundamentally from the generalized Dirichlet distribution (GED) [26]. Although both can be formulated into mixture models, they can not be reduced to each other. GED emphasizes correlation modeling in compositional data, while BMDD provides flexible modeling of componentwise bimodality. Furthermore, BMDD has more parameters than GED and defines a substantially more complex distribution.

## 2.2. True composition estimation for microbiome composition data

The posterior means from our model can be used to estimate the true compositions, upon which downstream statistical analyses can be performed. In this section, we present simulation studies that compare the performance of competing methods to recover the true compositions of microbiome data. We compared our method with mbDenoise and mbImpute, the methods for microbiome data imputation, and also SAVER, scImpute, and ALRA, the methods for scRNA-seq data. Furthermore, we compared our method to a related approach, DMM [27], which uses a Dirichlet multinomial mixture distribution for probabilistic modeling of microbiome data. Unlike BMDD, DMM assumes multiple mixtures of Dirichlet distributions on the full compositional vector rather than on each compositional component. We set the number of Dirichlet mixtures to 10 in our studies. We also included four naive strategies: do nothing (naive–1), add a pseudocount of 1 to all counts (naive–2), replace zeros with 0.5 (naive–3), and impute zeros by $N_i/(\max_{k:W_{k,j}=0} N_k)$ (see the definitions of $N_i$ and $W_{k,j}$ in Section 4.2) for the $j$th taxon in the $i$th sample (naive–4) as used in [28].

To perform a comprehensive evaluation, we considered multiple scenarios. We started by generating data using the BMDD model (denoted as setting S1; see Section A2.1 of the S1 File for details), where the parameters were estimated by fitting BMDD to the COMBO dataset from a cross-sectional study of the diet effect on stool microbiome composition [29]. In our evaluation, we simulated $n = 80$ samples and $m = 100$ taxa, reflecting typical microbiome abundance data at the genus level. Based on the true compositions and total counts, multinomial models were used to generate the observed counts.

For our method, the composition estimate is taken as the posterior mean of the true composition. To thoroughly assess the accuracy of different methods in estimating the true composition, we employ fifteen evaluation metrics, which can be broadly divided into three categories: (i) general similarity measures between two composition matrices, including mean squared error, sample-wise distance and taxon-wise distance; (ii) preservation of sample-wise properties, including Shannon's index, Simpson's index, Bray-Curtis dissimilarity, Kullback-Leibler divergence, Jensen-Shannon divergence, and Hellinger distance; (iii) preservation of taxon-wise properties, including the Gini coefficient, pair of mean and standard deviation, coefficient of variation, Kolmogorov–Smirnov distance, Wasserstein distance, and pairwise taxon-to-taxon correlation. Smaller values of these metrics indicate better performance. Specific formulations of these metrics are provided in Section A2.2 of the S1 File.

Fig 1c presents the results of setting S1 in terms of percentage improvement compared to the respective method. Detailed values are provided in Table A1 of the S1 File. BMDD outperforms competing methods in most evaluation metrics: of the 15 metrics, BMDD ranks first in 12 metrics, second in 2 metrics, and third in the remaining metrics (Table A1). The SAVER method, which considers the count sampling variability and the correlations among taxa, performs very well and achieves the second-best in most of the evaluation metrics. Other methods are substantially less optimal, and oftentimes, the performance could be inferior to the naive methods.

We are also interested in studying the robustness of our method under model misspecification, that is, when the data are not generated according to our BMDD model. As BMDD does not explicitly model the correlations among taxa, we are particularly interested in its performance when the data present such correlations. To achieve this, we conducted experiments in which the data was generated by different models (gamma, log-normal, Poisson, and negative binomial) with correlation structures. We use S2–S5 to denote these settings with the four distributions, respectively. Additionally, we performed simulations where data were generated from non-parametric models, using four real datasets (CDI, IBD, RA, SMOKE) as

the templates. We use S6–S9 to denote these non-parametric settings with the four datasets, respectively. Details of these settings can be found in Section A2.1 of the S1 File.

Under parametric models with correlations (settings S2–S5), BMDD and SAVER continue to outperform other methods (see Fig 2). In general, BMDD outperforms SAVER under the gamma, Poisson, and negative binomial distributions. Under the log-normal distribution, SAVER performs slightly better than BMDD. These results suggest that BMDD is robust to the correlation structure among taxa and model misspecification. Under the nonparametric models (settings S6–S9), the naive methods perform much better than under the parametric settings (Fig 2), which is not unexpected as the observed composition was not much different from the true composition due to the data-generation mechanism. Nevertheless, both the BMDD and the SAVER perform similarly to the naive methods, with BMDD exhibiting some advantages over the SAVER and naive methods. This further highlights the robustness of our BMDD approach.

The BMDD method employs the mean-field variational EM algorithm to estimate the parameters, which is computationally efficient and can be scalable to larger numbers of taxa. Fig 3 shows the execution time (measured in seconds per iteration) of the BMDD algorithm performed on the data generated from the setting S1. We observe that the BMDD algorithm can complete computations (several tens of iterations for most situations) within minutes for moderate sample size (e.g., 200) and number of taxa (e.g., 500) and within hours for a large number of taxa (e.g., 3000).

## 2.3. BMDD-based multiple imputation for differential abundance analysis of microbiome data

Differential abundance analysis is a central statistical task of microbiome data analysis, with the aim of identifying microbial taxa whose abundance covaries with a phenotype of interest in medical studies [30]. The identified taxa can provide insights into the etiology of the disease and can potentially be used as biomarkers for disease prevention, diagnosis, and treatment [1]. Log-linear models have been widely used in differential abundance analysis of microbiome data due to their simplicity, computational efficiency, and biological interpretability. The recently developed differential abundance analysis tools, ANCOMBC [31], MaAsLin [32], and LinDA [28], are all based on log-linear models. However, log-linear models require estimating the underlying true compositions and imputing the zeros first. One common approach involves adding a pseudocount, such as 0.5 or 1, to all counts before calculating the composition. The approach is ad hoc, and different pseudocounts could sometimes lead to very different results [33]. As there is much uncertainty in estimating the underlying composition, particularly for those zero counts, a single imputation or a single point estimate may not be sufficient, reducing the robustness of log-linear models. We thus hypothesize that multiple imputations based on posterior samples can significantly enhance the robustness of log-linear models by effectively accounting for uncertainty. Previously, we developed the LinDA method for differential abundance analysis with compositional bias correction [28]. One advantage of LinDA is that it can be applied to analyze correlated microbiome data using linear mixed-effects models. As the multiple imputed compositions can be regarded as repeated measurements, we can apply LinDA-LMM conveniently to integrate multiple imputations, thus accounting for the uncertainty in composition estimation. In the following, we conduct simulation studies and real data applications to compare the performance of original LinDA (which uses the pseudocount approach), LinDA with the posterior samples from our method (denoted as LinDA-BMDD), and LinDA with the posterior samples from SAVER (denoted as LinDA-SAVER) in terms of FDR control and power.

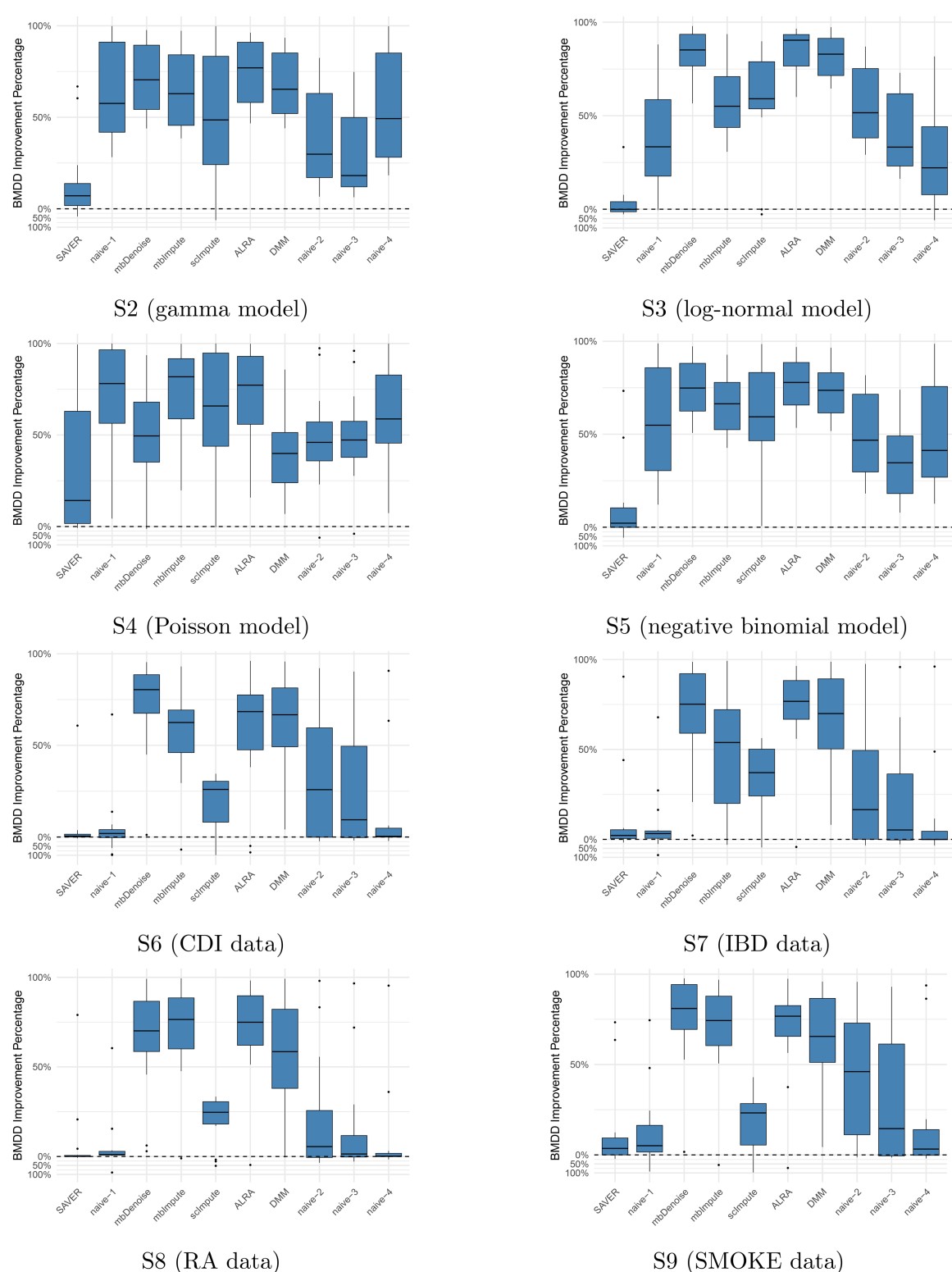

**Fig 2. Imputation performance comparison of BMDD with competing methods across 15 distance metrics between the estimated and true composition matrices under settings S2–S9.** The performance improvement is expressed as the percentage reduction in distance metrics relative to the respective method. Formula for the values above zero (the dashed horizontal line): $\frac{\text{SAVER}-\text{BMDD}}{\text{SAVER}} \times 100\%$, the distance reduction of BMDD relative to SAVER; Formula for the values below zero $\frac{\text{BMDD}-\text{SAVER}}{\text{BMDD}} \times 100\%$, the distance reduction of SAVER relative to BMDD. Likewise for other competing methods.

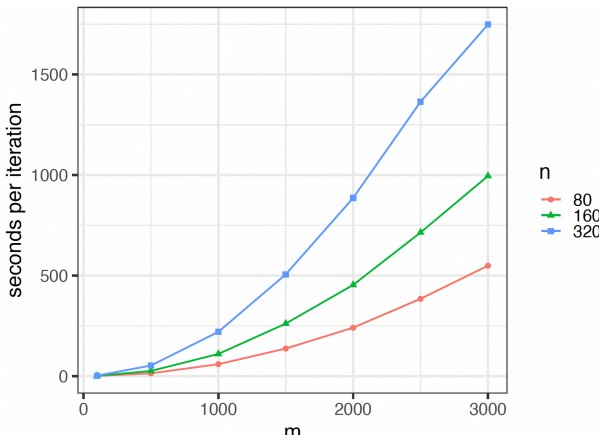

**Fig 3. Execution time of the BMDD algorithm.** (R version: 4.3.1 (2023-06-16); Platform: aarch64-apple-darwin20; CPU: Apple M2 Max; Memory: 32 GB). The y-axis represents the time (in seconds) that the BMDD algorithm needs to finish one iteration.

We employed similar simulation setups as those used in the LinDA paper [28]. Specifically, we included the setups with a log-normal distribution for absolute abundance, a gamma distribution for absolute abundance, and a negative binomial distribution for observed abundance (setups S0, S3, and S7 in [28], respectively). In addition, we added a setup with a Poisson distribution for the observed abundance in this study. Details of data generation can be found in Section A3.1 of the S1 File. We considered different configurations for the number of taxa and sample size: $m = 50, 200, 500$ and $n = 50, 200$. We adopted a moderate signal density (10% differential taxa) and signal strength such that we could see the power difference between methods.

We used 100 posterior samples of the true composition matrix for both LinDA-BMDD and LinDA-SAVER, so the input sample size for these two methods was $100n$. Under the gamma model, LinDA-BMDD controls the FDR across settings, with nonsignificant FDR inflation when the number of taxa is small (Fig 4a). In contrast, LinDA and LinDA-SAVER suffer from more serious FDR inflation in many settings, especially when the number of taxa is small. In terms of power, LinDA-BMDD has a slight power loss compared to the original LinDA, but the loss is not substantial, indicating that the increased robustness retains most of the statistical power. Under the log-normal model (Fig 5), LinDA-SAVER has the worst FDR control, with inflation worsening as the number of taxa increases. LinDA and LinDA-BMDD control the FDR overall, with only slight FDR inflation when the number of taxa is small. Under the Poisson and negative binomial (Fig 5), LinDA-BMDD again shows much better FDR control than LinDA-SAVER and LinDA.

We reason that the inferior FDR control by LinDA-SAVER could be due to "information diffusion", as zeros were imputed based on information from other taxa based on the underlying correlation structure. When there are many differential taxa, the differential signals may diffuse to those non-differential ones, increasing the false positive rate. Therefore, applying LinDA-SAVER is not recommended for differential abundance analysis, even if it is apparently highly powerful in some settings.

Overall, when the model is correctly specified for LinDA (that is, log-normal), the original LinDA performs well, and there is no significant benefit in FDR control with the use of the multiple imputation approach. However, when the model is misspecified, LinDA with BMDD

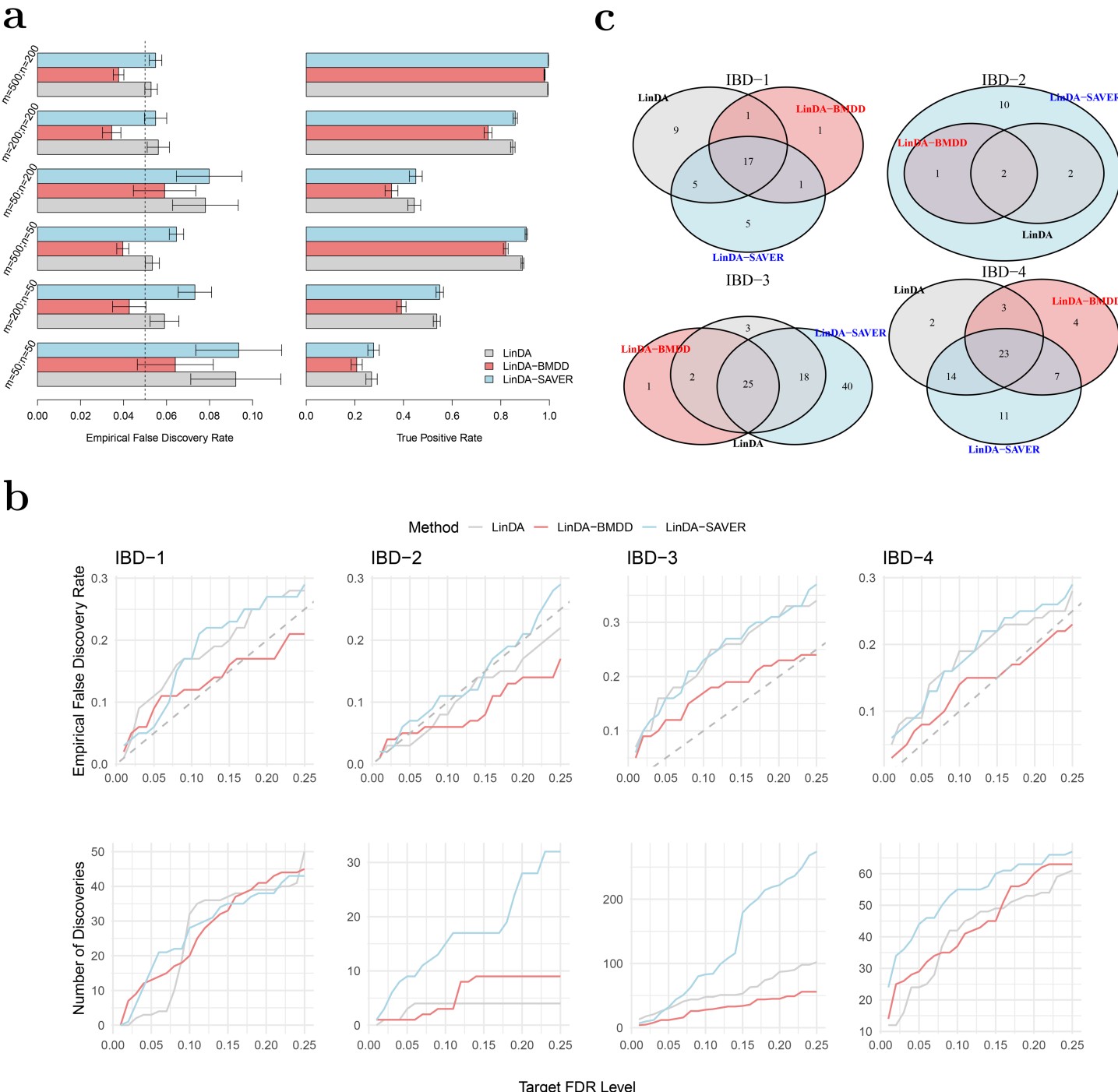

**Fig 4. Performance of BMDD in differential abundance analysis of microbiome data. a**: Simulation results for differential abundance analysis under the gamma model. Empirical false discovery rates and true positive rates were averaged over 500 simulation runs. Error bars represent the 95% confidence intervals and the dashed vertical line indicates the target FDR level of 0.05. **b**: (Bottom) Number of discoveries vs. target FDR level for the real datasets; (Top) Empirical FDR vs. target FDR level for the shuffled real datasets. The dashed gray line represents the diagonal. The results were averaged over 100 simulation runs. **c**: Overlaps of differential taxa with target FDR level of 0.1 for the four real datasets, IBD-1, IBD-2, IBD-3 and IBD-4.

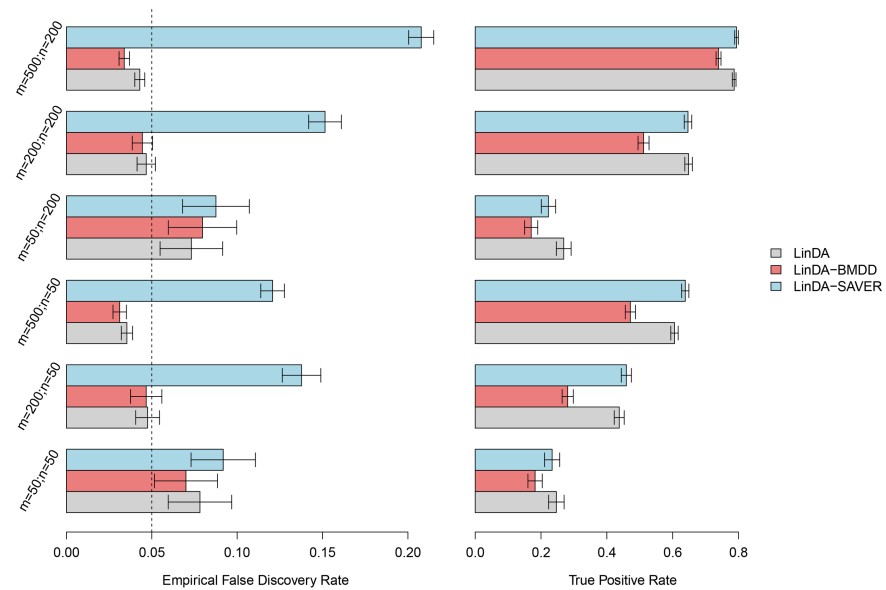

Log-normal model

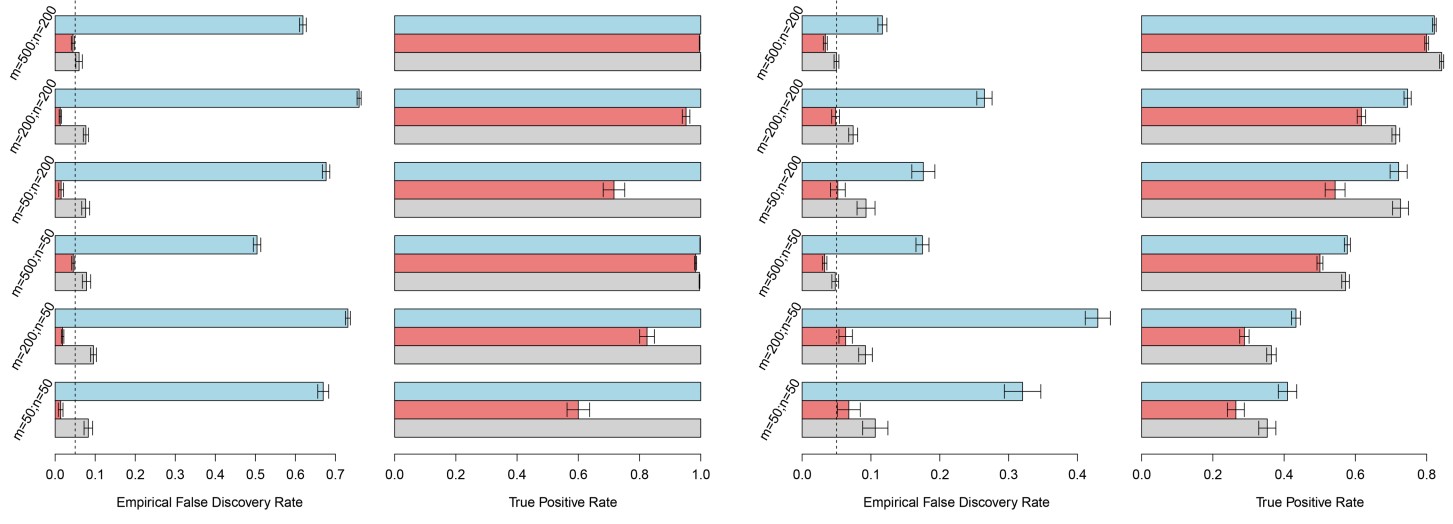

Poisson model                                  Negative binomial model

**Fig 5. Simulation results for differential abundance analysis under the log-normal, Poisson, and negative-binomial models, respectively.** Error bars represent the 95% confidence intervals and the dashed vertical line indicates the target FDR level of 0.05.

imputation significantly reduces the FDR inflation with only slight power loss. These simulation studies confirm that integrating the posterior inference using BMDD in LinDA leads to a more robust testing procedure by taking into account the uncertainty of the underlying true abundance under zero counts.

Next, we applied the three competing methods to four real datasets from case-control gut microbiome studies of inflammatory bowel disease (IBD). The four genus-level abundance datasets were retrieved from the MicrobiomeHD database [34]. We excluded samples with

fewer than 1000 read counts and taxa, which appear in less than 20% of the samples. We used winsorization at quantile 0.97 to reduce the impact of potential outliers as recommended in [35]. The characteristics of the filtered datasets are shown in Table A2 of the S1 File. We compared the number of discoveries of LinDA, LinDA-BMDD, and LinDA-SAVER at different FDR levels (0.01–0.25) and studied their overlap patterns at the target FDR of 0.1. As the ground truth is unknown, the assessment of the FDR control performance is difficult. However, we could at least evaluate the FDR control under the global null, where we disrupt the signals by shuffling the group labels. The empirical FDR was then calculated as the percentage of repetitions (random group label shuffling) that made any discoveries.

From Fig 4b, we can see that, in general, the patterns are consistent with the simulation studies. LinDA-BMDD significantly improves the FDR control over the original LinDA and LinDA-SAVER. The power of LinDA-BMDD is comparable to or slightly less than the original LinDA, echoing the simulation results. LinDA-SAVER has substantially more power than LinDA-BMDD for two datasets (IBD-2 and IBD-3), but the increased power could be due to its inefficient FDR control due to "information diffusion" discussed in the simulation results.

The overlap analysis (Fig 4c) reveals that most of the discoveries made by LinDA-BMDD at 10% FDR were also recovered by LinDA and/or LinDA-SAVER. In contrast, many discoveries made by LinDA-SAVER were not found by the other two methods. Those taxa recovered by LinDA-BMDD were also supported by previous literature. For example, in IBD-1, all three methods made similar numbers of discoveries, while only LinDA-BMDD offered adequate FDR control under the global null based on shuffled datasets. At 10% FDR, LinDA-BMDD, LinDA-SAVER, and LinDA identified 20, 28, and 32 differential taxa, respectively. The differential taxa identified by LinDA-BMDD showed a large overlap with LinDA-SAVER and LinDA. Nineteen taxa were also identified by either or both of the other methods. Only one taxon was unique to LinDA-BMDD. In contrast, LinDA-SAVER and LinDA had 9 and 5 method-specific taxa. The taxon identified only by LinDA-BMDD belonged to Clostridium XlVa of the Lachnospiraceae family. Lachnospiraceae, including Clostridium clusters IV and XIVa, were previously implicated in IBD [36]. Detailed studies of the results for IBD-2, IBD-3 and IBD-4 can be found in Section A3.3 of the S1 File.

To see if the increased robustness of differential abundance analysis is limited to LinDA, we further tested BMDD on ANCOM-BC [31,33], one of the most popular differential abundance analysis methods. We compared the performance of the original ANCOM-BC to ANCOMBC-BMDD and ANCOMBC-SAVAER. Since ANCOM-BC is computationally much slower, we used 20 posterior samples of the true composition matrix for ANCOMBC-BMDD and ANCOMBC-SAVER. The results on the number of discoveries and empirical FDR at different FDR levels are presented in Fig A2 of the S1 File. We observed that the original ANCOM-BC failed to control the FDR effectively: its empirical FDRs across different datasets and FDR levels were almost equal to 1, indicating that it always made some rejections when the group labels were shuffled. Interestingly, with the multiple imputation approach, using either the posterior samples from BMDD or SAVER, the FDRs were conservatively controlled. We note that in the newer version ANCOM-BC (ANCOM-BC2) [33], the authors recommended a sensitivity analysis based on different pseudocount additions. Using this strategy, the authors demonstrated significantly improved robustness of ANCOM-BC with more accurate FDR control. In comparison, our method proposed a more principled way to handle zeros without the need for sensitivity analysis.

## 3. Discussion

In this study, we proposed a Bayesian approach to estimate the true composition of the microbiome from the observed sequencing count data. We modeled the underlying true abundance using a mixture of gamma distributions for each taxonomic component, leading to a bimodal Dirichlet distribution for the taxonomic compositions. Using mean-field approximations for posteriors, we developed an efficient variational EM algorithm to obtain the estimates. Our results show that BMDD provides the best overall estimate of true compositions compared to existing methods for microbiome data based on extensive simulations. We also show that the proposed multiple imputation approach using posterior samples significantly improves the robustness of differential abundance analysis methods such as LinDA and ANCOM-BC.

The SAVER method, developed for single-cell sequencing data, also shows competitive performance. However, the constructions of the two methods are very different. SAVER employs a Poisson distribution with a gamma prior to modeling the counts and explores the relationship between genes in the same cell/sample (pooling information across genes), while BMDD utilizes a gamma mixture model for the counts of each taxon (pooling information across samples) and operates on the resulting bimodal Dirichlet distribution for the compositions. Thus, it is possible to combine the strengths of both methods to further improve the imputation performance. However, in differential abundance analysis, borrowing information from correlated taxa can lead to inflated false positive rates due to "information diffusion" from those truly differential taxa.

One drawback of BMDD is its limited ability to model the inter-taxa correlations. However, since the BMDD-based imputation method mainly borrows information from the abundance distribution of the same taxon in other samples, ignoring correlations among taxa is not expected to seriously affect its imputation performance. As suggested by the simulation results in Fig 2, BMDD model appears robust to the correlation structure among taxa.

Although BMDD assumes bimodality, it can still fit unimodal data (Fig 1a) because the added flexibility from more parameters can improve the fit. When the marginal distribution is beta-distributed, BMDD is overparameterized and the parameters are not identifiable; however, the underlying composition estimates remain accurate. We evaluated robustness under model misspecification and in nonparametric settings that mimic real data, and BMDD was generally resilient to deviations from bimodality. Regarding convergence, a formal proof for the variational approach is not yet available, but in practice, the posterior means were stable across taxa and across multiple initializations. BMDD's performance improves with greater sequencing depth and fewer taxa, as higher depth reduces sampling noise and aids recovery of the true composition, whereas increasing the number of taxa at fixed depth yields sparser counts and lower estimation efficiency.

We used 100 posterior samples because results varied little across repetitions. Although increasing the number of samples can further reduce Monte Carlo error, the incremental benefit is small. In practice, we recommend choosing a posterior sample size that yields stable results across runs. In the differential abundance analysis context, we used a mixed-effects model to incorporate multiple posterior samples. Alternatively, one may also analyze each posterior sample separately with a differential abundance method of choice (e.g., LinDA, ANCOM-BC, LOCOM [37]), extract the coefficient estimates and variances, and then combine results using Rubin's rules. We envision that multiple imputation using BMDD will be useful in other application settings such as community-wide testing, clustering, and prediction [38–40]. How to integrate multiple imputations efficiently in these statistical tasks warrants further investigation. Although our method was motivated by the analysis of microbiome data, it can be applied in principle to other zero-inflated compositional data analyses,

such as scRNA-seq data. However, rigorous benchmarking studies must be performed before BMDD can be recommended for other applications.

Our method can be potentially extended to accommodate both sample covariates and phylogeny among taxa using a mixture regression framework, as outlined in Section A4 of the S1 File, to further improve the efficiency of the imputation. Furthermore, through the regression framework, we can incorporate mixed effects to account for correlations among the samples arising from repeated measurement or longitudinal study of the microbiome. As this adds another layer of complexity, more computationally efficient methods are needed. Moreover, its robustness in the presence of misspecified covariates or incorrect phylogeny requires further study. We leave this as our future research direction.

Beyond its statistical applications, the BMDD model may provide insights into the underlying biological properties of the microbiome. For example, the inferred $\pi_j, \alpha_{j,0}$ and $\alpha_{j,1}$ values could be used to classify taxa. Those taxa with large $\pi_j$ and $\alpha_{j,1}$ values usually represent core taxa, which are observed stably across samples, while taxa with small $\pi_j$ and large $\alpha_{j,1}$ values are typical of transient taxa, which grow in a condition-dependent manner. In particular, the parameter $\pi_j$ can capture the bimodality of a given taxon, indicating whether its abundance follows a consistent pattern across samples or displays distinct patterns that may reflect ecological transitions (e.g., bloom dynamics [41], keystone switching [42]). Thus, examining the pattern of estimated $\pi_j$ may lead to interesting ecological findings.

## 4. Materials and methods

We develop a new probabilistic model for zero-inflated microbiome data. In particular, we assume the observed counts are generated from a multinomial distribution with the true composition following a BMDD to capture the variation of the composition data.

### 4.1. Motivation and formulation of BMDD

We first state a basic fact regarding the Dirichlet distribution, which inspires our formulation of BMDD. Let $Y_j$ be generated independently from the Gamma distribution with the shape parameter $\alpha_j > 0$ and scale parameter $\rho > 0$, that is, $\text{Gamma}(\alpha_j, \rho)$ for $1 \le j \le m$. Setting $V = \sum_{j=1}^{m} Y_j$, we have $(Y_1/V, \ldots, Y_m/V) \sim \text{Dirichlet}(\alpha_1, \ldots, \alpha_m)$. Motivated by this basic fact and the intention to capture the bimodal shape of taxa distribution, we consider a two-component mixture distribution for each $Y_j$, that is,

$$Y_j | \delta_j \sim (1 - \delta_j)\text{Gamma}(\alpha_{j,0}, \rho) + \delta_j\text{Gamma}(\alpha_{j,1}, \rho),$$

where $\delta_j \sim \text{Bernoulli}(\pi_j)$ independently over $j$, with success probability $0 \le \pi_j < 1$. Conditional on $(\delta_1, \ldots, \delta_m)$, one has

$$(X_1, \ldots, X_m) := (Y_1/V, \ldots, Y_m/V) \sim \text{Dirichlet}(\alpha_{1,\delta_1}, \ldots, \alpha_{m,\delta_m}),$$

with $X_j = Y_j/V$, which is independent of the choice of $\rho$. We name the above probability model the BiModal Dirichlet Distribution (BMDD). More formally, BMDD can be defined through the following hierarchical model:

$$(X_1, \ldots, X_m) \sim \text{Dirichlet}(\theta_1, \ldots, \theta_m),$$

$$\theta_j | \delta_j \overset{\text{ind}}{\sim} (1 - \delta_j)\omega(\alpha_{j,0}) + \delta_j\omega(\alpha_{j,1}),$$

$$\delta_j \overset{\text{ind}}{\sim} \text{Bernoulli}(\pi_j),.$$

where $\omega(x)$ denotes a point mass at $x$.

## 4.2. The probabilistic model

Let $W_{i,j}$ represent the observed read count of taxon $j$ in individual $i$. For the $i$th individual, the total count of all taxa, $N_i$, is determined by the sequencing depth and DNA materials. Let $X_i = (X_{i,1}, \dots, X_{i,m})$ denote the unobserved true composition for the $i$th individual. Given $N_i$, it is natural to model the stratified count data over $m$ taxa as a multinomial distribution

$$f\left(W_{i,j}, 1 \le j \le m | N_i\right) = \frac{N_i!}{\prod_{j=1}^{m} W_{i,j}!} \prod_{j=1}^{m} X_{i,j}^{W_{i,j}}, \quad 1 \le i \le n,$$

where $N_i = \sum_{j=1}^{m} W_{i,j}$. Assuming that $\{X_i\}_{i=1}^{n}$ are i.i.d samples generated from the BMDD, then we have the following hierarchical model:

$$\begin{aligned}
W_{i,1}, \dots, W_{i,m} | X_i &\sim \text{Multinomial}(X_{i,1}, \dots, X_{i,m}), \\
X_{i,1}, \dots, X_{i,m} | \theta_i &\sim \text{Dirichlet}(\theta_{i1}, \dots, \theta_{im}), \\
\theta_{ij} | \delta_{ij} &\overset{\text{ind}}{\sim} \delta_{ij}\omega(\alpha_{j,1}) + (1 - \delta_{ij})\omega(\alpha_{j,0}), \quad 1 \le j \le m, \\
\delta_{ij} &\overset{\text{ind}}{\sim} \text{Bernoulli}(\pi_j), \quad 1 \le j \le m,
\end{aligned} \tag{1}$$

where $\theta_i = (\theta_{i1}, \dots, \theta_{im})$. We summarize the BMDD structure in a schematic diagram (Fig 6). The goal is to estimate the hyperparameters $(\pi, \alpha)$ in the BMDD as well as the posterior distribution of $X$ with $X = (X_1, \dots, X_n)$, $\pi = (\pi_1, \dots, \pi_m)$ and $\alpha = (\alpha_{j,k})_{1 \le j \le m, k=0,1}$.

**Remark 2.** Our model (1) can be extended to incorporate optional metadata, including sample covariates and taxon phylogeny. The details can be found in Section A4 of the S1 File.

## 4.3. Variational inference

Let $W = (W_1, \dots, W_n)$ with $W_i = (W_{i,1}, \dots, W_{i,m})$. The joint likelihood of $(W, X)$ is given by

$$p(W, X | \pi, \alpha) = p(W | X) p(X | \pi, \alpha).$$

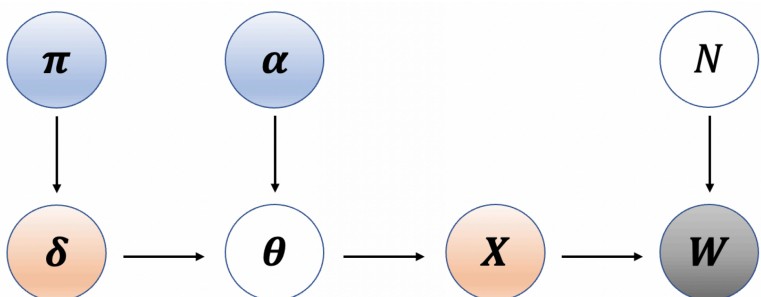

**Fig 6. The BMDD model.** $\pi$ and $\alpha$ are the model hyperparameters, $\delta$ represents the unobserved mode assignment variable governed by $\pi$, $\theta$ is the Dirichlet parameter ydetermined by $\delta$ and $\alpha$, $X$ represents the unobserved true composition generated by Dirichlet ($\theta$), and $W$ are the observed count data generated based on $X$ and a known sequencing depth $N$.

The major challenge is the computational intractability of the density

$$p(X_i|\pi,\alpha) = \sum_{\delta_{i1}=0}^{1} \cdots \sum_{\delta_{im}=0}^{1} \prod_{j=1}^{m} (1-\pi_j)^{1-\delta_{ij}} \pi_j^{\delta_{ij}} f(X_i|\alpha_{1,\delta_{i1}}, \ldots, \alpha_{m,\delta_{im}})$$

with $f(x|\alpha_1, \ldots, \alpha_m)$ denoting the density of Dirichlet$(\alpha_1, \ldots, \alpha_m)$.

To overcome this difficulty, we consider the variational frequentist estimate based on the mean-field approximation for the posterior. Specifically, we consider a class of distributions of the form

$$g(X|\beta)q(\delta|\gamma) = \prod_{i=1}^{n} g(X_i|\beta_i) \prod_{i=1}^{n} \prod_{j=1}^{m} q(\delta_{ij}|\gamma_{ij})$$

where $\delta = (\delta_{ij})_{1\leq i\leq n, 1\leq j\leq m}$, $\gamma = (\gamma_{ij})_{1\leq i\leq n, 1\leq j\leq m}$ with $0 \leq \gamma_{ij} < 1$, $\beta = (\beta_1, \ldots, \beta_n)$ with $\beta_i = (\beta_{i1}, \ldots, \beta_{im})$, $\beta_{ij} > 0$, $g(\cdot|\beta_i)$ represents the density function of Dirichlet $(\beta_{i1}, \ldots, \beta_{im})$ and $q(\cdot|\gamma)$ denotes the probability mass function of Bernoulli$(\gamma)$. Here $g(X|\beta)q(\delta|\gamma)$ is a variational distribution that serves as a surrogate for the posterior distribution $p(X, \delta|W, \pi, \alpha)$.

To motivate our algorithm, we note that

$$\log p(W|\pi,\alpha) = E_{(X,\delta)\sim gq}[\log p(W|\pi,\alpha)] = E_{(X,\delta)\sim gq}\left[\log \frac{p(W,X,\delta|\pi,\alpha)}{p(X,\delta|W,\pi,\alpha)}\right]$$

$$= E_{(X,\delta)\sim gq}\left[\log \frac{p(W,X,\delta|\pi,\alpha)}{g(X|\beta)q(\delta|\gamma)}\right] + E_{(X,\delta)\sim gq}\left[\log \frac{g(X|\beta)q(\delta|\gamma)}{p(X,\delta|W,\pi,\alpha)}\right]$$

$$= L(\beta,\gamma,\pi,\alpha) + D(g(X|\beta)q(\delta|\gamma)\|p(X,\delta|W,\pi,\alpha)),$$

where $L(\beta,\gamma,\pi,\alpha)$ is called the evidence lower bound (ELBO) in variational Bayes, and $D(\cdot\|\cdot)$ is the Kullback-Leibler (KL) divergence of the approximation from the true posterior. Therefore, maximizing the ELBO is equivalent to minimizing the KL divergence. We propose a variational EM algorithm to estimate the unknown parameters, which involves iterations of the following two steps:

1. E-step: fixing $(\alpha,\pi)$, update the parameter $(\beta,\gamma)$ in the mean field approximation through the coordinate ascent mean-field variational inference [43];
2. M-step: fixing $(\beta,\gamma)$, update the hyperparameters $(\alpha,\pi)$ by maximizing the ELBO.

Below, we describe these two steps in detail.

### 4.4. Variational EM algorithm

**E-step: update the mean-field approximation.** Given $\pi$ and $\alpha$, we aim to update the mean-field distribution by maximizing the ELBO. We employ one of the most commonly used algorithms for solving this optimization problem, the coordinate ascent variational inference (CAVI). According to CAVI, we deduce the optimal form of $g(X_i|\beta_i)$,

$$\log g(X_i|\beta_i) = \log p(W_i|X_i) + E_{\delta\sim q}[\log p(X_i|\delta_i,\alpha)] + \text{const}$$

$$= \sum_{j=1}^{m} W_{i,j} \log(X_{i,j}) + \sum_{j=1}^{m} \left(\{\gamma_{ij}\alpha_{j,1} + (1-\gamma_{ij})\alpha_{j,0} - 1\} \log X_{i,j}\right) + \text{const}.$$

 

It thus implies that $\beta_{ij} = W_{i,j} + \gamma_{ij}\alpha_{j,1} + (1-\gamma_{ij})\alpha_{j,0}$ for $1 \leq j \leq m$. On the other hand, the optimal form of $q(\delta_{ij}|\gamma_{ij})$ is given by

$$\log q(\delta_{ij}|\gamma_{ij}) = E_{(X,\delta)\sim gq}\left[\log p(X_i|\delta_i,\alpha)|\delta_{ij}\right] + \log p(\delta_{ij}|\pi_j) + \text{const}$$

$$= E_{(X,\delta)\sim gq}\left[\log\left(\Gamma\left(\sum_{k=1}^{m}\alpha_{k,\delta_{ik}}\right)\prod_{k=1}^{m}\frac{X_{i,k}^{\alpha_{k,\delta_{ik}}-1}}{\Gamma(\alpha_{k,\delta_{ik}})}\right)\Bigg|\delta_{ij}\right]$$

$$+ \delta_{ij}\log(\pi_j) + (1-\delta_{ij})\log(1-\pi_j) + \text{const}$$

$$= h_{ij}(\delta_{ij}) + \delta_{ij}\log(\pi_j) + (1-\delta_{ij})\log(1-\pi_j) + \text{const},$$

where $h_{ij}(\delta_{ij}) = Q(\delta_{ij}) + (\alpha_{j,\delta_{ij}}-1)\{\psi(\beta_{ij}) - \psi(\sum_{k=1}^{m}\beta_{ik})\} - \log\Gamma(\alpha_{j,\delta_{ij}})$ with $0\,Q(\delta_{ij}) = E_{\delta\sim q}[\log\Gamma(\sum_{k=1}^{m}\alpha_{k,\delta_{ik}})|\delta_{ij}]$, and we have used the fact that $E_{X\sim g}[\log X_{i,j}] = \psi(\beta_{ij}) - \psi(\sum_{j=1}^{m}\beta_{ij})$, where $\psi$ denotes the digamma function. Therefore, we obtain

$$\gamma_{ij} = \frac{\pi_j\exp\left(h_{ij}(1)\right)}{\pi_j\exp\left(h_{ij}(1)\right) + (1-\pi_j)\exp\left(h_{ij}(0)\right)}.$$

One can approximate the expectation in $Q(\delta_{ij})$ via the Monte Carlo approach. Note that $h_{ij}$ depends on $\gamma_{ik}$ for $k \neq j$. One needs to iterate the above updates across $j = 1, 2, \ldots, m$ until convergence. The iteration can be done in a parallel fashion across $i = 1, 2, \ldots, n$.

**Remark 3.** The Monte Carlo approximation for $Q(\delta_{ij})$ can be computationally very intensive. To reduce the computation cost, we consider the following approximate approach based on Jensen's inequality. Using the fact that the log gamma function is convex on the positive real line, we have

$$E_{\delta\sim q}\left[\log\Gamma\left(\sum_{k=1}^{m}\alpha_{k,\delta_{ik}}\right)\Bigg|\delta_{ij}\right] \geq \log\Gamma\left(\alpha_{j,\delta_{ij}} + \sum_{k\neq j}\{\gamma_{ik}\alpha_{k,1} + (1-\gamma_{ik})\alpha_{k,0}\}\right).$$

Maximizing this lower bound leads to the optimal form of $q$ given by $h_{ij}$ with the term $Q(\delta_{ij})$ in $h_{ij}$ being replaced by $\log\Gamma(\alpha_{j,\delta_{ij}} + \sum_{k\neq j}\{\gamma_{ik}\alpha_{k,1} + (1-\gamma_{ik})\alpha_{k,0}\})$. This approach is computationally much faster than the Monte Carlo approximation. In addition, with Monte Carlo estimates, the dependence of the objective function on the parameters is stochastic and lacks a closed-form expression, which introduces instability into the optimization. In contrast, with the lower bound, the dependence on the parameters is explicit and deterministic, leading to a more stable and standard optimization procedure.

**M-step: update the hyperparameters.** Next, we update the hyperparameters by optimizing the ELBO with respect to $(\pi, \alpha)$:

$$E_{(X,\delta)\sim gq}\left[\log p(W, X, \delta|\pi, \alpha)\right]$$

$$= E_{(X,\delta)\sim gq}\left[\log p(W|X)\right] + E_{(X,\delta)\sim gq}\left[\log p(X|\delta,\alpha)\right] + E_{(X,\delta)\sim gq}\left[\log p(\delta|\pi)\right]$$

$$= E_{(X,\delta)\sim gq}\left[\log\left(\prod_{i=1}^{n}\Gamma\left(\sum_{j=1}^{m}\alpha_{j,\delta_{ij}}\right)\prod_{j=1}^{m}\frac{X_{i,j}^{\alpha_{j,\delta_{ij}}-1}}{\Gamma(\alpha_{j,\delta_{ij}})}\right)\Bigg|\alpha\right]$$

$$+ \sum_{i=1}^{n}\sum_{j=1}^{m}\left\{\gamma_{ij}\log\pi_j + (1-\gamma_{ij})\log(1-\pi_j)\right\} + \text{const}$$

 

Again the term $E_{\delta \sim q}\left[\log \Gamma\left(\sum_{j=1}^{m} \alpha_{j,\delta_{ij}}\right)\right]$ is not tractable. We address this issue using the same trick as in Remark 3 and propose to update $\pi$ by letting $\pi_j = \frac{1}{n}\sum_{i=1}^{n} \gamma_{ij}$ and update $\alpha$ by maximizing the following objective function through the optimization function "nlminb" in the R software,

$$\sum_{i=1}^{n} \log \Gamma\left(\sum_{j=1}^{m}\left\{\gamma_{ij}\alpha_{j,1} + (1-\gamma_{ij})\alpha_{j,0}\right\}\right) + \sum_{i=1}^{n}\sum_{j=1}^{m}$$

$$\left[\left\{\gamma_{ij}\alpha_{j,1} + (1-\gamma_{ij})\alpha_{j,0} - 1\right\}\left\{\psi(\beta_{ij}) - \psi\left(\sum_{j=1}^{m}\beta_{ij}\right)\right\} - \gamma_{ij}\log\Gamma(\alpha_{j,1}) - (1-\gamma_{ij})\log\Gamma(\alpha_{j,0})\right].$$

## Supporting information

**S1 File. List of Legends.**

- **Fig A1. Posterior predictive check for the four example taxa from the American Gut Project.** The blue histogram shows the observed data, and the red histogram represents data generated by sampling from multinomial distributions parameterized with posterior samples of the compositions from BMDD. "0s" indicates the proportion of zeros, and "SD" denotes the standard deviation.
- **Fig A2. Performance of ANCOM-BC with BMDD on the four real datasets.** (Bottom) Number of discoveries vs. target FDR level for the real datasets; (Top) Empirical FDR vs. target FDR level for the shuffled real datasets. The dashed gray line represents the diagonal. The results were averaged over 100 simulation runs.
- **Fig A3. A simplified illustration of the phylogenetic tree.** ◯ denotes internal node while △ represents leaf node.
- **Table A1. Simulation results for true composition estimation under setting S1 (BMDD model).** The values outside the parentheses are estimates of the corresponding evaluation metrics achieved by each method (averaged over 10 simulation runs), and the values in parentheses are standard errors. Bold red and red values in each row indicate the two best methods for each evaluation metric.
- **Table A2. Characteristics of the real microbiome datasets.** The second to fifth columns respectively list the number of taxa, sample size, numbers of the controls and cases of each filtered dataset (prevalence ≥ 20%, library size ≥ 1000).
(PDF)

## Author contributions

**Conceptualization:** Jun Chen, Xianyang Zhang.

**Formal analysis:** Huijuan Zhou, Xianyang Zhang.

**Funding acquisition:** Jun Chen, Xianyang Zhang.

**Methodology:** Huijuan Zhou, Jun Chen, Xianyang Zhang.

**Software:** Huijuan Zhou.

**Supervision:** Jun Chen, Xianyang Zhang.

**Visualization:** Huijuan Zhou.

**Writing – original draft:** Huijuan Zhou.

**Writing – review & editing:** Jun Chen, Xianyang Zhang.

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
