## [Decision Letter · Decision Letter 0]

22 Jul 2025

PCOMPBIOL-D-25-00901

BMDD: A Probabilistic Framework for Accurate Imputation of Zero-inflated Microbiome Sequencing Data

PLOS Computational Biology

Dear Dr. Chen,

Thank you for submitting your manuscript to PLOS Computational Biology. After careful consideration, we feel that it has merit but does not fully meet PLOS Computational Biology's publication criteria as it currently stands. Therefore, we invite you to submit a revised version of the manuscript that addresses the points raised during the review process.

Please submit your revised manuscript within 60 days Sep 21 2025 11:59PM. If you will need more time than this to complete your revisions, please reply to this message or contact the journal office at ploscompbiol@plos.org. Please include the following items when submitting your revised manuscript:

We look forward to receiving your revised manuscript.

Kind regards,

Alberto J M Martin, Ph.D.

Academic Editor

PLOS Computational Biology

Ilya Ioshikhes

Section Editor

PLOS Computational Biology

**Additional Editor Comments:**

Before resubmitting your article, please address all major issues raised by both reviewers. In addition, ensure you improve the readability of the article so it becomes easier to read, specially mind final end users who might not be experts on the method. Keep this in mind also for the manual in the github repository

**Journal Requirements:**

At this stage, the following Authors/Authors require contributions: Huijuan Zhou, Jun Chen, and Xianyang Zhang. Please ensure that the full contributions of each author are acknowledged in the "Add/Edit/Remove Authors" section of our submission form.

6) Please send a completed 'Competing Interests' statement, including any COIs declared by your co-authors. If you have no competing interests to declare, please state "The authors have declared that no competing interests exist". Otherwise please declare all competing interests beginning with the statement "I have read the journal's policy and the authors of this manuscript have the following competing interests"

**Reviewers' comments:**

Reviewer's Responses to Questions

**Comments to the Authors:**

Reviewer #1: See attached

Reviewer #2: thanks for the nice job

I have couple of questions and couple of suggestions

1) How do you ensure identifiability between the two Gamma components in the mixture model?

Especially for taxa with very sparse signals, the difference between the two α parameters (α₀ and α₁) may be subtle. Could different parameter combinations produce similar outputs?

2) How stable and biologically interpretable is the πⱼ parameter across datasets?

Since πⱼ reflects the probability of the second (non-zero) mode, do you observe consistency in πⱼ for known opportunistic taxa or context-dependent microbes?

3) Though you state that the final composition is independent of ρ, does the choice of ρ affect inference stability or convergence in practice?

4) Have you performed posterior predictive checks to evaluate whether BMDD accurately reproduces multimodal structures in held-out taxa?

A visual or statistical PPC would build confidence in model fit and flexibility.

5) Could the BMDD framework be extended to incorporate temporal or longitudinal structure?

For microbiome studies with repeated measurements (e.g., interventions), accounting for temporal dynamics would be highly valuable.

6) Is the bimodal assumption flexible enough for cases with more than two ecological states?

Some taxa may have trimodal or multimodal behavior depending on niche availability or host status.

7) Can you connect the inferred bimodality to known ecological transitions (e.g., bloom dynamics, keystone switching)?

Showing such links would strengthen the biological impact of your model

these are the suggestions that the paper could benefit from:

1) Discuss how inferred πⱼ values could be used to stratify taxa (e.g., core vs. transient microbes).

2) While Figure 1b shows the generative process, a higher-level conceptual diagram summarizing assumptions and inference flow (akin to a graphical model) would help non-statistical readers

3) Especially for real data, show how the model captures bimodal or zero-inflated features—e.g., overlaid histograms or kernel density plots of observed vs. simulated abundance for selected taxa

4) Explicitly discuss:

When the bimodal assumption may fail

Whether the variational posterior converges reliably across taxa

How model performance scales with increasing taxa or deeper sequencing

5) It’s not clear if code for BMDD is publicly available. Publishing the pipeline (e.g., as an R or Python package) would vastly improve its adoption.

6) The multiple imputation framework is appealing. But it’s worth clarifying:

How many posterior samples are recommended in practice?

Is Rubin's rule applicable for any DA method, or just linear models?

Overall, I believe BMDD represents a compelling improvement over traditional Dirichlet and zero-inflated methods for modeling microbiome compositions.

all the best

**Have the authors made all data and (if applicable) computational code underlying the findings in their manuscript fully available?**

Reviewer #1: **No: **I did not find any simulation study in the given Github repo

Reviewer #2: **No: **I couldn't find the raw data and the code

PLOS authors have the option to publish the peer review history of their article (what does this mean?). If published, this will include your full peer review and any attached files.

Reviewer #1: **Yes: **Lukas Arnroth

Reviewer #2: No

**Figure resubmission:**
---

## [Decision Letter · Decision Letter 1]

3 Oct 2025

Dear Dr. Chen,

We are pleased to inform you that your manuscript 'BMDD: A Probabilistic Framework for Accurate Imputation of Zero-inflated Microbiome Sequencing Data' has been provisionally accepted for publication in PLOS Computational Biology.

Best regards,

Alberto J M Martin, Ph.D.

Academic Editor

PLOS Computational Biology

Ilya Ioshikhes

Section Editor

PLOS Computational Biology

Reviewer's Responses to Questions

**Comments to the Authors:**

Reviewer #1: You have answered all questions in a satisfying manner.

Minor comment:

I think you could move the section in the discussion on the GED ("BMDD differs fundamentally from the generalized Dirichlet distribution...") to section 2.1 (possibly as a remark) as I think it is more natural there.

Reviewer #2: Thanks for the comprehensive responses. I believe the paper is now readable and ready to be received by the community.

**Have the authors made all data and (if applicable) computational code underlying the findings in their manuscript fully available?**

Reviewer #1: Yes

Reviewer #2: Yes

PLOS authors have the option to publish the peer review history of their article (what does this mean?). If published, this will include your full peer review and any attached files.

Reviewer #1: **Yes: **Lukas Arnroth

Reviewer #2: No

---

## [Editor Report · Acceptance letter]

PCOMPBIOL-D-25-00901R1

BMDD: A Probabilistic Framework for Accurate Imputation of Zero-inflated Microbiome Sequencing Data

Dear Dr Chen,

I am pleased to inform you that your manuscript has been formally accepted for publication in PLOS Computational Biology. Your manuscript is now with our production department and you will be notified of the publication date in due course.

With kind regards,

Zsofia Freund
